# Semi-Supervised Reinforced Active Learning for Pulmonary Nodule Detection in Chest X-rays

**Sejin Park**[*]
Vuno Inc.
Seoul, South Korea
gnoses@vuno.co

**Woochan Hwang**[*]
Imperial College London
London, UK
woochan.hwang14@ic.ac.uk

**Kyu-Hwan Jung**
Vuno Inc.
Seoul, South Korea
khwan.jung@vuno.co

## Abstract

Machine learning applications in medical imaging are frequently limited by the lack of quality labeled data. While conventional active learning approaches have been able to reduce the labeling burden to some extent, the main difficulty was defining an effective sampling criteria. In this work, we propose a novel framework, *semi-supervised reinforced active learning*, which utilizes inverse reinforcement learning and an actor critic network to train a reward based active learning algorithm. This is an extension of the reinforced active learning formulation [1] to complex problems where direct rewards may be unavailable. The framework was tested on a U-Net segmentation network [2] for pulmonary nodules in chest X-rays. The proposed framework was able to achieve the same level of performance as the standard U-Net while using only 50% of the labeled data, demonstrating ability to effectively reduce the labeling burden.

## 1   Introduction

Machine learning applications in medical imaging face a common obstacle of obtaining quality labeled data. The integrity of most labels in medical imaging is inherently limited as many of them are created through natural language processing or classification based on the information from the PACS system. Though this may seem straightforward, the content of the medical report may not always be identical to the radiological findings. For example, a patient might have the final diagnosis of lung cancer in the report, but the PACS system may contain X-rays taken before any observable nodules or consolidations have formed. Furthermore, in images with multiple findings, the report may only include the representative finding while excluding the others.

Even in the few selective cases where expert-level segmentation labels are available, it is immensely time consuming and expensive to scale to the amount where supervised learning algorithms can be implemented effectively. This also confines the application of most supervised learning algorithms in the medical domain to problems with a clear financial return while neglecting many with a potentially huge clinical impact.

In such a limited environment where it is difficult to obtain enough X-ray data with validated labels, the logical approach is that of semi-supervised learning, where the aim is to utilize the vast amount of X-rays with no or unverified labels. The active learning style [3], which selects and labels a subset of the unlabeled dataset in an iterative way, is the currently established approach in this domain. In this paper, we propose using an advantage actor critic (A2C) network [4] to replace the conventional sampling methods of active learning algorithms. This is founded on the idea of a reward-driven active learning algorithm, which was first proposed as reinforced active learning formulation (RALF) [1] for object classification. The reward is defined by the performance of the network after being

---

[*]Equal contribution

1st Conference on Medical Imaging with Deep Learning (MIDL 2018), Amsterdam, The Netherlands.

fine tuned with the subset created by the actor. However, because the end performance of the base network is intractable during training, we take the semi-supervised reinforcement learning (SSRL) [5] approach and formalize the overall framework as semi-supervised reinforced active learning (SSRAL). This approach not only clarifies the weak link between sampling methods and performance of conventional active learning methods but also provides a stable learning method in the context of pulmonary nodule detection.

The main contribution is that, to the best of our knowledge, our work is the first implementation of the reinforced active learning approach in the medical imaging domain. Our work describes a novel way to reduce the labeling burden, which is one of the main bottlenecks in this field, by exploiting unlabeled data. Furthermore, our work improves on the reinforced active learning approach to accommodate more complex problems where the reward function may not be available by introducing semi-supervised reinforcement learning.

## 2  Background

Semi-supervised learning [6] is based on the desire to make use of unlabeled data in settings where labeled data is scarce but unlabeled data is available at scale. The intuition behind why it works however, may not be as clear. Zhu et al. [7] provides some insight by showing how semi-supervised learning occurs in humans. The intuition is that unlabeled data can provide useful information of the true distribution of the data and shift the decision boundary of the model, leading to improved generalization of classifiers. The potential of semi-supervised learning in reducing the labeling burden has been proven empirically [8,9,10] in several domains including segmentation tasks in medical imaging.

Most of such semi-supervised learning methods have been implemented in the active learning style [1], where a query is thrown at the user (or teacher model) to generate data, which will then be used for fine tuning of the model. More specifically, in the medical imaging domain, we can imagine a setting where we would want to utilize the large amount of unlabeled X-rays readily available for a certain task. In active learning, a query will be thrown to generate labels for a subset of unlabeled X-rays that will be incorporated into the training process. The popular methods [11,12,13] used for subsampling from the unlabeled set are as follows:

1. Uncertainty sampling: Omission of data with high uncertainty
2. Query by committee: Sampling through voting of multiple pre-trained supervised learning models
3. Expected model change: Selection of data points that drastically change the current model
4. Balance exploration and exploitation: Solving the contextual bandit problem of exploration and selection of the data points

Although these conventional methods have proven effective in multiple occasions, there is still a lot of room for improvement. In all of the above cases, the process of sampling and validation of performance are independent. Hence, no direct feedback is given to the sampling algorithm to improve over iterations. Because the choice of sampling method leads to drastic changes in performance, the lack of a direct feedback complicates the experiment. Furthermore, such methods require long training periods which limits its application on large image sets.

One way of overcoming such limitations of conventional sampling methods may be to integrate reinforcement learning (RL) in the sampling process. Reinforcement learning is a branch of machine learning inspired from behavioral neuroscience where the agent recognizes the current state and selects the action or sequence of actions that will maximize the reward. Ebert et al. [1] first introduced the RALF algorithm based on this approach by formulation of the sampling criteria as a Markov decision process. Unlike RALF which took a model-free Q-learning approach for training, we implement a modified version of the advantage actor critic (A2C) [4] method in our model. By combining the policy gradient approach (actor) with the value based approach (critic), the actor critic method provides a better convergence rate compared to most of the older algorithms in high dimensional space [14].

As the training of supervised learning algorithms rely on the quality of the label, the reinforcement learning algorithms rely on the quality of the reward. However, in the case of deep neural architectures

used in the medical imaging domain, the final validation performance is intractable during the training process. The iterative training performance, which is available, may be highly unstable and lead to non-convergence if used as a direct reward even with sophisticated policy gradient methods. To tackle this issue, we consider the semi-supervised reinforcement learning (SSRL) [5] problem, which is when an agent must perform RL under a setting where the reward function is known for only some of the cases. SSRL was formalized previously in the benchmark settings popular in robotics. To clarify overlapping terminology, SSRL is being used to implement a reinforced active learning style semi-supervised algorithm. Therefore, to avoid confusion, the proposed framework will be referred to as semi-supervised reinforced active learning (SSRAL).

In SSRAL, we use inverse reinforcement learning (IRL) [15,16] to implement a function approximation of the reward based on expert demonstration provided by the environment. This is different from the IRL implemented in the S3G [5] algorithm, which infers the reward function from a policy trained on a small set of labeled cases. By using this reward function in joint with the in-training validation performance, we were able to stabilize the learning curve, which is a key step in implementing reinforced active learning algorithms in complex real world problems like medical imaging. We will show the potential of the SSRAL framework in reducing the labeling burden in the medical imaging domain by experimentation on the pulmonary nodule detection task in chest X-rays.

## 3 Methods

### 3.1 Data Collection and Preprocessing

The chest X-rays (PA view) used were collected between 2013 and 2015 from Asan Medical Center (Seoul, South Korea). The dataset is comprised of 931 images with pixel labels (1007 nodules) and 2986 images without labels. The labels were created by consensus from three board certified radiologists with 10 to 25 years of experience. The total of 3917 radiographs used were anonymized and reviewed by the internal review board.

During the pre-processing stage, per image histogram equalization was used to mitigate the variance of intensity amongst the radiographs. In our model, data augmentation (rotation, random crop, resize, intensity and contrast noise) had no significant effect on training and validation accuracy. Therefore, no augmentations were applied to our data as they increased the training time with no significant benefit.

### 3.2 Semi-Supervised Reinforced Active Learning

Let a Markov Decision Process (MDP) be defined as $M = \{S, A, T, R, \gamma\}$, where S denotes the state space, A denotes the action space, T is the transition dynamics, R is the reward and $\gamma$ is the discount factor. In the given active learning setting, the state space of the MDP is the output of the trained model and the action is subsampling informative cases with generated labels. A sample is considered informative if the state space is below the negative threshold, in which case will be labeled normal, or above the positive threshold, where the region of interest will be considered a true nodule. The optimal policy $\pi^*$ is the sampling method that provides the highest expected reward.

Although there should be no restriction to the architecture of the trained model that we can take an SSRAL approach, a U-Net [2] segmentation network, which provides state-of-the-art performance in the medical imaging domain, was used. Because the final performance of the U-Net is intractable during the training iterations, maximum margin IRL [16] was used to create an approximation of the reward function based on expert demonstration. This was used in joint with the validation accuracy to update the A2C network.

Using the given MDP definition, we will formalize how the agent interacts with the environment to exploit unlabeled data based on feedback from the environment. The training of the proposed model is divided in two phases like most semi-supervised algorithms.

**Phase 1 (Supervised Learning):** A U-Net like model is trained in four different settings, each using 25%, 50%, 75% and 100% of the available labeled data. These will be compared after phase 2, which is the SSRAL training phase, to observe how robust the method is with less labeled data. The trained model becomes the environment that interacts with the agent in phase 2. The state space will be

defined as the logit output of the model to prevent unnecessary loss of information caused by the final sigmoid activation layer.

At the end of phase 1, a set of expert demonstrations is created from the environment using the state space of the labeled training data and its original labels. The reward function $R^*$ is approximated from the experts behavior via maximum margin IRL.

$$\mathbb{E}[\sum_{t=0}^{\infty} \gamma^t R(s_t)^* | \pi^*] \geq \mathbb{E}[\sum_{t=0}^{\infty} \gamma^t R(s_t)^* | \pi] \quad \forall \pi$$

The approximated reward function is used by the critic in phase 2 to evaluate the policy. No iterative updates are performed on the reward function itself.

**Phase 2 (SSRAL):** A subset of the unlabeled data is created based on the current policy, which starts with random initialization, and used to fine tune the segmentation network. The resulting validation accuracy $r$ at this stage is considered the long term reward. The policy and value function of the A2C network is trained to maximize the short term reward $R^*(s)$ provided by the IRL trained in phase 1 and the long term reward $r$. The use of $R^*(s)$ stabilizes the learning that was unstable when only based on $r$. The value function is calculated based on temporal difference (TD) methods [17] using both $R^*(s)$ and $r$.

$$\delta^{\pi\theta} = r \cdot R^*(s) + \gamma V^{\pi\theta}(s') - V^{\pi\theta}(s)$$

The iterative update of the actor is based on the policy gradient,

$$\theta = \theta + \alpha \nabla_\theta \log \pi_\theta(s,a) Q_w(s,a)$$

where the true TD error is an unbiased estimate of the advantage function. Note that unlike many of the previous active learning algorithms, this method describes a clear link between the final performance and the iterative updates.

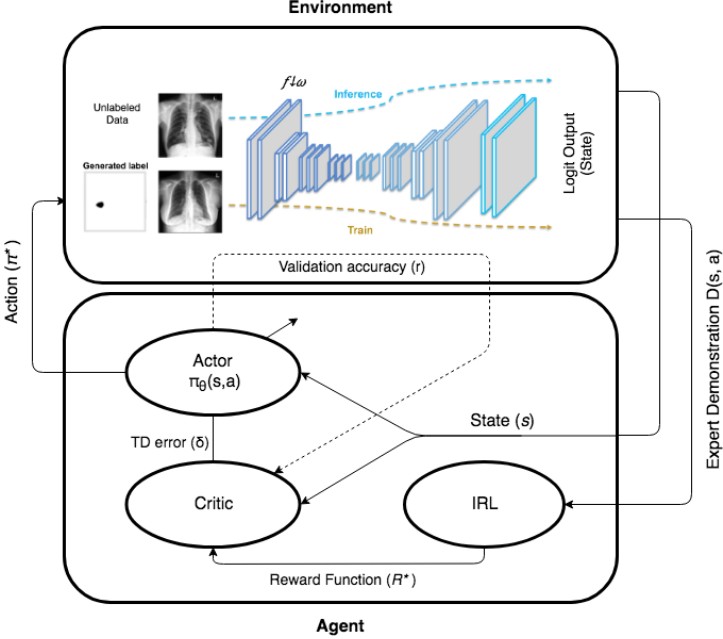

Figure 1: Semi-Supervised Reinforced Active Learning (SSRAL) framework

# 4 Results

In our evaluation, we compare (1) the performance of the model network between phase 1 and phase 2 to test the improvement following active learning and (2) the performance of the SSRAL framework

Table 1: Five fold validation results of the proposed framework using different proportions of the available labeled data. The result of t-Tests comparing phase 1 (Standard U-Net) and phase 2 (SSRAL) F1 scores are given as p-values.

| Labeled data used | Phase 1 | | | Phase 2 | | | t-Test (F1) |
|---|---|---|---|---|---|---|---|
| | F1 | Sensitivity | FPs/Img | F1 | Sensitivity | FPs/Img | |
| 25% | $0.738 \pm 0.015$ | 0.732 | 0.675 | $0.764 \pm 0.027$ | 0.780 | 0.742 | <0.001 |
| 50% | $0.745 \pm 0.018$ | 0.772 | 0.508 | $0.802 \pm 0.014$ | 0.829 | 0.312 | <0.001 |
| 75% | $0.794 \pm 0.023$ | 0.822 | 0.534 | $0.821 \pm 0.019$ | 0.865 | 0.262 | <0.001 |
| 100% | $0.812 \pm 0.014$ | 0.856 | 0.342 | $0.848 \pm 0.022$ | 0.887 | 0.252 | <0.001 |

with varying degree of labeled data use to test how effective the framework is in reducing the labeling burden.

The five fold validation performance of phase 1 and phase 2 using different proportions of the available labeled data is shown in Table 1. The p-value indicates the results of the t-Test performed on the f1 score of respective phase 1 and phase 2 performance. The result demonstrates that the phase 2 of the training improves the network performance throughout.

The phase 1 performance achieved using 100% of the labeled data can be considered the maximum performance of the original U-Net segmentation network. Figure 2 shows that with the SSRAL framework, we can achieve the same level of performance (F1 score) with only 50% of the labeled data (p-value < 0.05). Figure 3 shows a sample where the SSRAL framework was able to detect nodules that were missed by the standard U-Net trained with 50% of the available labeled data.

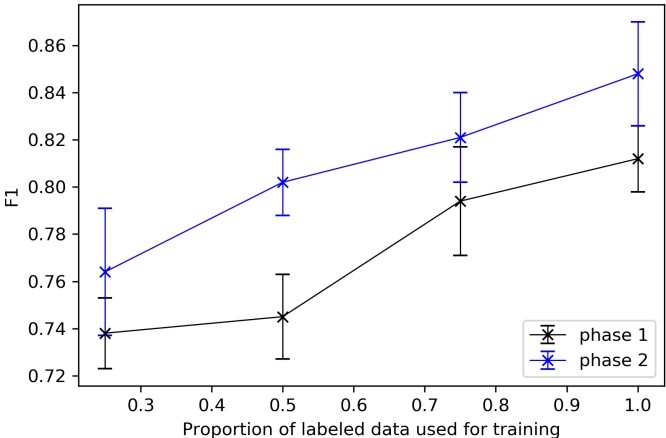

Figure 2: Performance (F1 score) of the proposed framework using different proportions of the available labeled data. Error bar is given by standard deviation.

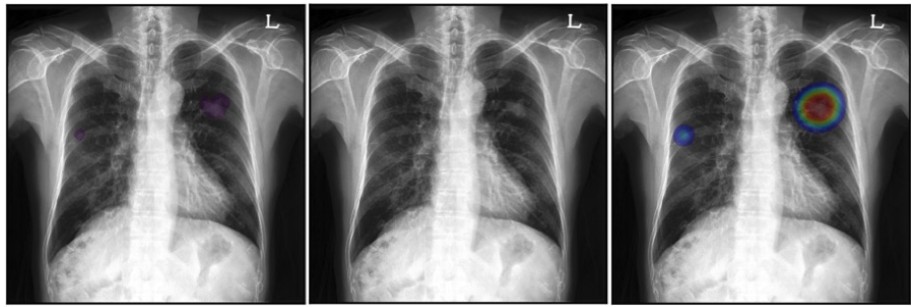

Figure 3: Inference result of the model trained using 50% of available labeled data on sample data. (Left) Ground truth, (Middle) Phase 1: Standard U-Net, (Right) Phase 2: SSRAL

# 5 Conclusion and Future Work

We presented a form of active learning algorithm that receives performance feedback and is robust enough to be implemented on complex real world problems, where the available reward may be highly abstract and sparse. To learn in such settings, our algorithm derives an approximated reward function through IRL and updates the A2C network based on a combination of long and short term rewards. Because this framework takes intuition from semi-supervised reinforcement learning and reinforced active learning formulation, we formalized this as semi-supervised reinforced active learning.

Our evaluation on the pulmonary nodule detection task in chest X-rays using the U-Net segmentation network showed that our approach can effectively leverage unlabeled data to improve performance of deep neural networks. In the specific dataset and task, we were able to reduce the labeling burden to 50% while maintaining performance. Though further experimentation on different datasets and tasks is necessary to gauge the true value of this framework, our results show great potential. A particularly interesting area where this framework may be useful is the multi-center adaptation problem. With some consideration in adjusting the reward function, our approach could be used to generalized a pretrained algorithm to settings with different patient characteristics without any further labeled data.

## Acknowledgements

This study was supported by the Industrial Strategic Technology Development Program of the Ministry of Trade, Industry & Energy (10072064) in the Republic of Korea.

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
