# OpenReview forum: "Semi-Supervised Reinforced Active Learning for Pulmonary Nodule Detection in Chest X-rays"
_MIDL.amsterdam/2018/Conference — Submitted to MIDL 2018_

### Review · AnonReviewer2 · 2018-04-27
**Review for the SSRAL Pulmonary Nodule Detection in Chest X-rays**

**Rating:** 2
**Confidence:** 2

**Review:**

This paper introduces a new active learning (AL) methodology for the detection of pulmonary nodules from chest x-ray.  The approach is heavily based on RALF [1], which is an AL method that dynamically combines exploration and exploitation in order to select the most informative samples to be annotated.  The main contribution of this work is to replace the Q-learning in [1] by the advantage actor critic (A2C) network [4].  The paper also claims that the paper is the first to use reinforced active learning in medical imaging, which is not quite precise, for example, see the paper: Gutiérrez B, Peter L, Klein T, Wachinger C. A Multi-Armed Bandit to Smartly Select a Training Set from Big Medical Data. In International Conference on Medical Image Computing and Computer-Assisted Intervention 2017 Sep 10 (pp. 38-45).  Experiments are based on a five-fold cross validation experiment with 931 images with pixel labels (1007 nodules) and 2986 images without labels.  F1, sensitivity and FP results are shown comparing the proposed method and a supervised training approach, showing that  SSRAL works better using different proportions of the available labelled data.

This is an interesting submission, with potentially interesting results, but not clearly written.  I believe that in its current state, it is not possible to re-implement the proposed technique, and the understanding of the paper is only possible if the reader is aware of RALF [1].  It is expected that papers are in general self-contained, but that is clearly not the case for this submission.  In addition, given that this is not the first method to apply reinforced active learning in medical imaging, I expect that the results show a comparison with the baseline approach.

Minor issues:
- Results in Table 1 are not clear.  The first column indicates the proportions of the available labelled data used - is that the initial proportion?  When phase 2 runs, additional images will be labelled, so the comparison should be done as a function of the labelled images.  In addition, it is important to show the results with a random selection of training images to label.
- When you use 100% of the available labelled data in phase 1, how is the active learning used?  In particular, in Fig. 2, how come the results with 100% for phase 1 and phase 2 different?

**Special Issue:**

No

---

> ### Comment · ~Sejin_Park1 · 2018-05-01
> **Comparision to RALF and MAB paper in Miccai 2017**
>
> Thank you for your kind review, Though we fully respect your opinion, there were a few points that we wanted to address regarding comparison to existing active learning methodologies in medical imaging.
> 1. Comparison to RALF
> The network proposed as RALF is using Q-learning to replace the classification sampling strategy. They implemented a guided initialization phase to stabilize the unstable training due to the lack of an initial transition map. However, because our problem setting is segmentation, we needed to train a sampling network and a label generation network simultaneously. Stabilizing the training for our network is much more complex and can not be solved given the ideas in the RALF paper. This is why the use of A2C and IRL is key in our study and why there was no direct performance comparison.
> 2. “A Multi-Armed Bandit to Smartly Select a Training Set from Big Medical Data”
> The main motivation of this paper is to sample the most useful data to reduce training time based on the meta-information(volumetric information from freesurfer, sex, age, etc). However, our approach is leveraging the unlabelled dataset based on only the pixel information directly. Therefore, though it maybe true that AL methods have been used in the medical imaging domain, to the best of our knowledge, none of them have been based on the images themselves.
>
> Regarding to minor issues you mentioned,
> - The first column of table 1 indicates partial label count we used. For 1 rows (25% proportion), we didn't use another 75% of labeled data at all in phase 1 and 2. In phase 2, unlabeled data is only used commoly for all proportions.
> - In 100 % proportion, we used all labeled data in phase 1. Then we used unlabeled data in phase 2.
> - Additionally, unlabeled data is extrememly noisy data which consist of nodule findings, normal patients and other disease findings (effusion, penumonia, and so on)

---

### Review · AnonReviewer3 · 2018-05-09
**Unclear description of the model architecture, training and evaluation procedure. Comparison to baseline methods missing.**

**Rating:** 1
**Confidence:** 2

**Review:**

The authors formulate a semi-supervised active learning task as a reinforcement learning problem to reduce the required amount of labelled data to train a segmentation network. Unfortunately the approach is presented in a way that is impossible to follow. It remains unclear which amount of labelled data, unlabelled data, requests for additional labels, and the validation set is used at which part of the training process. The different parts in figure 1 are not explained, and the figure is not referenced from the main text. It is also not clear, how the whole approach was quantitatively evaluated and why an active learning approach can achieve a higher performance than the underlying model that is trained with 100% of the training data. Finally comparisons to baseline methods (i.e. other methods to select further training data) are missing.


**Special Issue:**

No

---

### Review · AnonReviewer1 · 2018-05-09
**Interesting method, needs more comparison with other active learning methods**

**Rating:** 2
**Confidence:** 2

**Review:**

In this paper, the authors proposed a semi-supervised reinforced active learning (SSRAL) method with an A2C network for sampling the unlabeled data. The targeting problem is very important. It will help general reader to understand the paper if the authors can present more information of RALF and A2C. The experiments are a little bit confusing. I understand it after reading the comment from the authors to reviewer2.

Here are major comments:
1.	The main issue with this paper is there is no quantitative comparison with other active learning methods. All the comparisons are against their own methods. The experimental numbers are mostly a reflection of common sense: more data (labeled or unlabeled) can generally help.
2.	How did the author create the “generated labels” for the unlabeled data? Explanation of the A2C or RALF may help the reader understand how the annotation effort is saved, as well as how these unlabeled data is utilized.
3.	What are the failure cases? Those cases having good segmentation in Phase 1 but not so well in Phase 2?
4.	What is the loss function for Phase 1? Is there any unbalanced issue?


**Special Issue:**

No

---

### Decision · Program_Chairs · 2018-05-15
**Paper92 Acceptance Decision**

Reject